# Proton-Pump Inhibitors Suppress T Cell Response by Shifting Intracellular Zinc Distribution

**DOI:** 10.3390/ijms24021191

**Published:** 2023-01-07

**Authors:** Wenlei Liu, Jana Jakobs, Lothar Rink

**Affiliations:** Institute of Immunology, Faculty of Medicine, RWTH Aachen University Hospital, Pauwelsstraße 30, 52074 Aachen, Germany

**Keywords:** proton-pump inhibitors, T cells, zinc, Zip8, CREM, pCREB, IL-2, IFN-γ

## Abstract

Proton-pump inhibitors (PPI), e.g., omeprazole or pantoprazole, are the most widely used drugs for various gastrointestinal diseases. However, more and more side effects, especially an increased risk of infections, have been reported in recent years. The underlying mechanism has still not yet been fully uncovered. Hence, in this study, we analyzed the T cell response after treatment with pantoprazole in vitro. Pantoprazole preincubation reduced the production and secretion of interferon (IFN)-γ and interleukin (IL)-2 after the T cells were activated with phytohemagglutinin (PHA)-L or toxic shock syndrome toxin-1 (TSST-1). Moreover, a lower zinc concentration in the cytoplasm and a higher concentration in the lysosomes were observed in the pantoprazole-treated group compared to the untreated group. We also tested the expression of the zinc transporter Zrt- and Irt-like protein (Zip)8, which is located in the lysosomal membrane and plays a key role in regulating intracellular zinc distribution after T cell activation. Pantoprazole reduced the expression of Zip8. Furthermore, we measured the expression of cAMP-responsive element modulator (CREM) α, which directly suppresses the expression of IL-2, and the expression of the phosphorylated cAMP response element-binding protein (pCREB), which can promote the expression of IFN-γ. The expression of CREMα was dramatically increased, and different isoforms appeared, whereas the expression of pCREB was downregulated after the T cells were treated with pantoprazole. In conclusion, pantoprazole downregulates IFN-γ and IL-2 expression by regulating the expression of Zip8 and pCREB or CREMα, respectively.

## 1. Introduction

Proton-pump inhibitors (PPIs) are widely used in the treatment of various gastrointestinal diseases such as peptic ulcer, gastroesophageal reflux disease, Zollinger–Ellison syndrome, and upper gastrointestinal bleeding [1,2]. Pantoprazole and omeprazole are most commonly used in medical treatment at the moment. PPIs covalently bind to the acid pump enzyme H^+^ /K^+^-ATPase and they irreversibly inactivate it [3]. By acting on the final step of gastric acid secretion, these drugs effectively inhibit gastric acid secretion regardless of the presence of other factors. However, more and more side effects have been reported in recent years [2,4,5], especially their suppressive effect on the immune system [6,7,8]. Meanwhile, in vivo and in vitro experiments have demonstrated that the status of intracellular free zinc has an important regulatory influence on the human immune system [9,10,11,12].

Even though there are no reliable biomarkers to accurately screen the status of zinc in vivo, it has been widely accepted for years that the status of free zinc of lysosomes and cytoplasm can be assessed using FluoZin-3 AM and Zinpyr-1 staining, respectively [13,14]. Using these two different staining techniques, the distribution of intracellular zinc under different conditions can be studied.

Zip8 is important for the maintenance of intracellular zinc homeostasis in different cell types [15,16]. A 20–25-fold increase in mRNA levels after T cell activation [17,18] makes it an important factor in the post-activation response of T cells. The knockdown of Zip8 by siRNA decreases IFN-γ production, and conversely, an increased expression of Zip8 increases the expression of IFN-γ [18,19].

Zinc is an essential cofactor for T cell generation, maturation, differentiation, and proliferation [20,21]. Zinc deficiency leads to thymic atrophy and decreased thymulin production, resulting in decreased T cell proliferation and increased apoptosis. The expression of the anti-apoptotic factor B cell lymphoma-2 is reduced, and caspase activity is increased during zinc deficiency, leading to a decrease in the number of immature CD4/CD8 double-positive T cells [20].

In addition, in zinc deficiency, the expression and secretion of different cytokines after T cell activation is altered. Recent studies by our group have shown that zinc deficiency leads to reduced IL-2 transcription and secretion by increasing the expression of CREMα, a negative regulator of IL-2 transcription [11]. In zinc-deficient elderly patients, zinc supplementation reverses the negative effect of zinc deficiency, leading to low CREMα expression and thus normal IL-2 transcription [22].

In addition, IFN-γ expression is altered in different zinc conditions. It was shown that a high concentration of cytoplasm-free zinc reduces the activity of the phosphatase calcineurin (CN), thereby sustaining the phosphorylation of the transcription factor CREB, which in turn increases IFN-γ expression [18].

The aim of the study was to elucidate the immunosuppressive effect of PPIs and their effect on zinc homeostasis. We hypothesized that PPIs alter zinc homeostasis and thereby affects cytokine expression.

## 2. Results

### 2.1. Effect of PPI on Cytokine Production

To investigate the immunosuppressive effect of PPIs, we examined cytokine production after T cell activation. IFN-γ is an indicator of T helper cell 1 activation and is essential for immunity against intracellular pathogens and tumors [23], and IL-2 is important for T cell proliferation and differentiation [18,24,25]. We preincubated peripheral blood mononuclear cells (PBMCs) with 75 μM pantoprazole, one kind of PPI, for 48 h and then stimulated the cells with 1 μg/mL PHA or 50 ng/mL TSST for 24 h or 48 h. An at least 2-fold decrease in both IFN-γ (Figure 1A–D) and IL-2 (Figure 1E–H) expression was observed in the PPI-treated group after stimulation with PHA or TSST, with the largest decrease of almost 13-fold after 24 h of TSST stimulation (Figure 1G).

### 2.2. Effect of PPIs on the Distribution of Intracellular-Free Zinc

It was previously shown that IFN-γ and IL-2 expression are regulated by intracellular zinc [9,11,18,22]; thus, we next examined the distribution of intracellular-free zinc in PPI-treated PBMC. The lysosome-free zinc concentration was determined by FluoZin-3 AM, and the cytoplasm-free zinc concentration was determined by Zinpyr-1 staining. In the PBMC, after being treated with PPI, a time-dependent increase in lysosome-free zinc (Figure 2A,C) and a decrease in cytoplasm-free zinc (Figure 2B,D) were observed. These results suggested that PPIs may cause a shift in the distribution of intracellular zinc.

### 2.3. The Influence of PPIs on the Expression of Zip8

In the following experiment, we investigated the redistribution of intracellular zinc after PPI treatment. Zip8 is located in the lysosomal membrane, and it transports zinc from the lysosome into the cytoplasm and has been shown to be important in IFN-γ expression [18]. Thus, we investigated the expression of Zip8 in the PBMCs in the control and the PPI-treated group by Western blot analysis. A significant decrease in the expression of Zip8 was observed after 24 h (Figure 3A,B) and 48 h (Figure 3A,C) of treatment with PPIs.

### 2.4. PPI Treatment Increased the Number of CREM Isoform

In order to investigate the reduced expression of IL-2 in the PPI-treated PBMC, we examined CREMα, which has been shown to inhibit IL-2 transcription in zinc deficiency [11,22]. The human CREM gene consists of 20 exons [26,27], and alternative splicing results in many different isoforms that have different, even opposite, effects on target gene expression. After PBMCs were treated with pantoprazole, the expression of CREMα was investigated by Western blot analysis. Preincubation of the PBMCs with pantoprazole upregulated the expression of total CREM (Figure 4A–C) and shifted the present isoforms of CREM (Figure 4D–I). The expression of CREM at 45 kD was downregulated, whereas at 80 kD and 100 kD it was upregulated. These results suggested that PPIs upregulate CREM and thereby negatively regulate IL-2 transcription.

### 2.5. The Influence of PPIs on the Expression of pCREB

To investigate the inhibitory effect of PPIs on the secretion of IFN-γ, we examined the transcription factor pCREB, which is highly expressed under high zinc concentrations and induces IFN-γ transcription [18]. The density of pCREB in the PBMCs after being treated with pantoprazole for 24 h (Figure 5A,B) or 48 h (Figure 5A,C) decreased in a time-dependent manner when compared to the control group. These results suggested that PPIs reduce IFN-γ expression by downregulating pCREB.

## 3. Discussion

PPIs are widely and even excessively used in gastrointestinal diseases [1,2]. In recent years, more and more side effects [2,4,8], especially a high rate of infection in patients [6,7,8], have been reported. PPIs have been suspected to be associated with different infections. In fact, *Clostridium difficile* infection (CDI) is the most common infection observed in PPI users [28]. In recent years, many studies and meta-analyses have revealed an association between PPI treatment and CDI [28,29]. This high infection rate is attributed to the higher transgastrointestinal resistance of the ingested spores and the alteration of the microbiota. In addition to CDI, many other infections such as *Salmonella*, *Campylobacter*, and community-acquired pneumonia occur but cannot be explained solely by the mechanisms just mentioned. A study from our laboratory also found that PPI treatment increases the risk of infection in patients undergoing cardiac surgery [6]. We also found that in vitro and in vivo pantoprazole treatment reduces the production of different pro-inflammatory cytokines and the function of polymorphonuclear cells. Therefore, a better understanding of the mechanisms of PPI-induced immunosuppression may contribute to a better understanding of PPI side effects.

The first step of this work was to confirm the inhibitory effect of PPIs on PBMCs. Therefore, we stimulated PPI-treated and untreated PBMCs with PHA [18,30] and TSST [31], respectively. PHA is a natural lectin that binds to sugars on glycosylated surface proteins and crosslinks the T cell receptor, causing cytokine release [32,33]. TSST is a bacterial superantigen that binds directly to the Vβ region of the T cell receptor and connects the TCR to the MHC protein, independent of a specific antigen [31]. This activates T cells non-specifically and triggers a cytokine storm [34].

For both T cell activation models, we found a reduced cytokine release in the PPI-treated cells. Interestingly, previous work showed that TSST-induced cytokine release is not sensitive to high zinc concentrations [31]. However, our results showed that the TSST-stimulated cells were sensitive to PPI treatment and thus to a zinc-deficient state, suggesting that a physiological zinc concentration is an important prerequisite for the activation of T cells by TSST.

Contrary to our results, it has also been reported that high zinc concentrations inhibit the production of IL-2 [35] expression, but we believe that there can be several explanations for this, for example, different experimental conditions and zinc concentrations.

Intracellular free zinc is necessary for different types of cells to perform their corresponding functions [36,37], especially in naive and mature T cells [20,21,38]. Therefore, we next tested the intracellular zinc distribution status after PPI treatment. A time-dependent decrease in cytoplasm-free zinc and an increase in intra-lysosomal-free zinc in the PPI-treated group was found when compared to the control group. We found that cytoplasm-free zinc was increased 24 h after PPI treatment, but it decreased after 48 h. However, not only did the cytoplasm-free zinc concentration in the PPI-treated group change but that in the untreated control group also gradually increased from 24 to 48 h.

We think that the gradual increase in cytoplasm-free zinc in the control group may have been a mild response to the in vitro environment and this response may have been due to the zinc transporter located on the cell membrane or the release of intracellular zinc bound to metallothioneins (MT). Still, this shifted intracellular zinc distribution could explain why the PPI-treated group produced lower amounts of cytokines after PHA or TSST stimulation.

To investigate the molecular mechanism of shifted intracellular-free zinc redistribution after PPI treatment, we measured the expression of Zip8, a membrane zinc transporter localized in the lysosomal membrane [39]. Zip8 is the most important zinc transporter that transports free zinc from the lysosome to the cytoplasm. When T cells were activated, Zip8 expression increased 20–25 fold, making it the strongest response, followed by Zip3 and Zip14 [18]. We found a time-dependent decrease in Zip8 expression in the PPI-treated group when compared to the control group. A previous study showed that the knockdown of Zip8 with siRNA reduced IFN-γ expression and secretion. In contrast, overexpression of Zip8 had the opposite effect, expressing and secreting more IFN-γ [18]. In our experiment, we did not measure other zinc transporters, but interestingly, we found that the reduction in cytokine secretion could not be reversed through zinc supplementation, both before and after PPI treatment (Liu et al. Institute of Immunology, RWTH Aachen University, Aachen, Germany, ELISA measurement). These results implied that intra-lysosomal zinc, but not extracellular free zinc, had a critical role in cytokine release.

The inhibition of cytokine production by the elevated expression of CREMα caused by a reduced intracellular zinc concentration was first proposed by our laboratory [11]. CREMα is expressed in most immune cells and is encoded into many different protein isoforms that have a wide range of effects on multiple signaling pathways and organ functions through post-transcriptional modifications [40,41]. For example, CREMα was found to play a key role as an epigenetic and transcriptional regulator in T lymphocytes [42]. A number of important selective splicing events occurred in the PPI-treated group compared to the untreated group. Although we did not investigate the specific role of these protein isoforms, we assumed that this alteration in regulatory protein isoforms was associated with altered cytokine secretion. Of course, more experiments are needed if we want to fully understand the PPI-induced changes in CREMα protein isoforms and their effects on cytokine production.

The phosphorylation status of CREB that is affected by intracellular zinc status [43] can be explained by the zinc-sensitive catalytic enzyme calcineurin (CN) [44], which promotes CREB dephosphorylation [45]. It has been shown that the activity of CN is inhibited at high zinc concentrations of 0.8–25 μM [18], resulting in an increase in pCREB and thus a higher IFN-γ transcription. This is consistent with our observation that in a zinc-deficient state less pCREB and less IFN-γ is expressed.

In this study, we focused on the zinc signaling pathway that is influenced by PPIs in immune cells. There might be some other ways that are involved in the immunosuppressive effect of PPIs. Therefore, more research on this topic is necessary.

In conclusion, the shifted distribution of intracellular free zinc was the result of the suppression of Zip8 in the PPI-treated PBMCs. The low cytoplasm-free zinc concentration led to a differential expression of CREMα isoforms and a reduced expression of pCREB, which ultimately resulted in the downregulation of IL-2 and IFN- γ (Figure 6).

## 4. Materials and Methods

### 4.1. Isolation of Human PBMC

Blood was obtained via venipuncture from healthy young volunteers with informed consent and ethics committee approval (RWTH Aachen University Hospital, document No. EK023/05). PBMCs were separated by Ficoll gradient centrifugation as described by our group [46]. In short, peripheral whole blood was diluted 1:2 with PBS (Sigma-Aldrich, Steinheim, Germany) and then put gently onto Ficoll. After centrifugation, cells in the interphase were collected and washed in PBS. In between, red blood cells were lysed with distilled water. Cells were resuspended in RPMI 1640 medium (Sigma-Aldrich, Germany) supplemented with 10% heat-inactivated fetal calf serum (FCS) (Bio&Sell, Nuremberg, Germany), 2 mM L-glutamine, 100 U/mL potassium penicillin, and 100 ug/mL streptomycin sulfate (all from Sigma-Aldrich). Cells were adjusted to concentrations indicated in the specific assays.

### 4.2. Cell Culture and PPI-Treated Models

The isolated human PBMCs were incubated at 37 °C in a humidified 5% CO_2_ atmosphere with or without pantoprazole in a final concentration of 75 μM and incubated for 24 h or 48 h. After the incubation with pantoprazole, PBMCs were collected to measure intracellular zinc concentration and the expression of Zip8, CREMα, and pCREB. After 48 h of incubation with pantoprazole, PBMCs were additionally activated with 1 μg/mL PHA-L or 50 ng/mL TSST for 24 h and 48 h. After 24 h or 48 h, the supernatants were collected to measure the expression of IFN-γ and IL-2.

### 4.3. Flow Cytometric Measurement of Intracellular Free Zn^2+^ with FluoZin-3 AM and Zinpyr-1

Isolated human PBMCs were incubated, as described before, with or without preincubation with pantoprazole. A total of 1 × 10^6^ cells per sample were incubated in 1 mL PBS for 30 min either with 1 µM FluoZin-3 AM (Invitrogen, Darmstadt, Germany) or 10 µM Zinpyr-1 (Chemodex, St. Gallen, SG, Switzerland) at 37°C in the dark. Afterward, cells were washed with 2 mL PBS and resuspended with 900 μL PBS. Samples were divided into 3 tubes. Tubes were incubated for 10 min at 37 °C with either TPEN (50 µM) to obtain minimal fluorescence, with ZnSO_4_/pyrithione (100 µM/5 µM) (all Sigma-Aldrich, Germany) to obtain maximal fluorescence, or were left untreated. Subsequent flow cytometry measurements were performed using FACSCalibur (BD, New Jersey, USA) and gated for analysis. Calculation of intracellular labile zinc was performed as described before using a dissociation constant, KD, of 8.9 nM for the FluoZin-3/Zn^2+^ complex and a KD of 0.7 nM for the Zinpyr-1/Zn^2+^ complex.

### 4.4. IFN-γ and IL-2 Quantification

Supernatants for IFN-γ and IL-2 determination were harvested from 1 × 10^6^ cells/mL. After activating the PBMCs as described above, supernatants were collected and stored at −20 °C until measurement. IFN-γ and IL-2 protein concentrations in the supernatants of the PPI-treated or untreated PBMCs were determined by OptEIA ELISA assay (BD, Germany) according to the manufacturer’s instructions with a detection limit of 4.7 pg/mL and 7.8 pg/mL, respectively. Only the incubation times of the standard and samples were adjusted to 2 h, and the incubation time of the IL-2 detection antibody was adjusted to 1 h; the incubation times of streptavidin–horse radish peroxidase (HRP) conjugate and subsequent substrate solution were adjusted to 18 min and 20 min, respectively. Results were measured by a Spark microplate reader (Tecan, Crailsheim, Germany).

### 4.5. Western Blot

Analysis of the following antibodies was conducted: Zip8 (protein tech), CREM C-2 (Santa Cruz Biotechnology, Germany), and pCREB (cell signaling). A total of 2 × 10^6^ isolated PBMCs were preincubated with pantoprazole or left untreated for two days and then were immediately prepared for Western blot analysis. For preparing the cells for Western blot analysis, cells were collected, centrifuged, and washed with 1 mL PBS. After that, the cells were resuspended in 100 μL lysis buffer (65 mM Tris–HCl [pH 6.8], 2% [*w*/*v*] SDS, 1 mM sodium orthovanadate, 26% [*v*/*v*] glycerol, 1% [*v*/*v*] β-mercaptoethanol, and 0.01% [*w*/*v*] bromphenol blue). A Vibra Cell sonicator (Sonics & Materials, Newtown, CT, USA) was used to lyse the cells. The lysed cells were heated for 5 min at 95 °C. A total of 30 μL of samples per lane was separated at 90 V for half an hour and then at 170 V at the end on 10% polyacrylamide gels for all antibodies used. Samples were blotted onto nitrocellulose membranes with a pore size of 0.45 μM at 300 mA for 1 h (GE Health-care Life Sciences, Boston, MA, USA). Then, membranes were blocked for 1 h in TBS-T (20 mM Tris [pH 7.6], 137 mM NaCl, and 0.1% [*v*/*v*] Tween 20) + 5% fat-free dry milk and washed in TBS-T afterwards. Subsequently, the membranes were incubated with the respective primary antibody overnight at 4 °C (CREM C2: 1/500 dilution in TBS-T + 5% fat-free dry milk; Zip8: 1/300 dilution in TBS-T + 5% FCS; pCREB: 1/1000 dilution in TBS-T + 5% FCS; β-actin: 1/2000 dilution in TBS-T + 5% fat-free dry milk). After washing again with TBS-T, the membranes were incubated with the respective secondary antibody, namely anti-mouse-HRP for CREM C-2 or anti-rabbit-HRP for β-actin, pCREB, and Zip8 for at least 2 h (both diluted 1/2000 in TBS-T + 5% fat-free dry milk; from Cell Signalling Technology, Danvers, MA, USA). Immunodetection was performed with LumiGlo Reagent (Cell Signalling Technology, USA) using LAS-3000 (Fujifilm Lifescience, Düsseldoff, Germany). Band density was analyzed with ImageJ software (Version 1.53k, National Institutes of Health, Bethesda MD, USA).

### 4.6. Statistical Analysis

The data are shown as the mean values ± SEM. Significance was analyzed by two-tailed Student’s *t*-test; *, **, and *** represent *p* ≤ 0.05, *p* ≤ 0.01, and *p* ≤ 0.001, respectively.

## Figures and Tables

**Figure 1 ijms-24-01191-f001:**
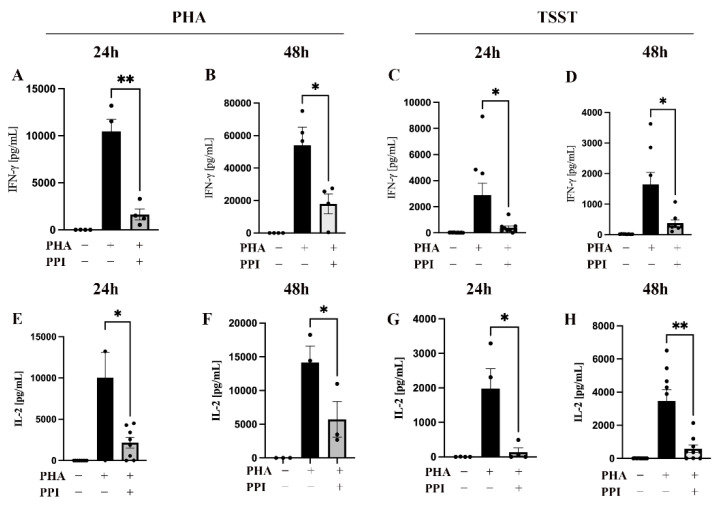
Influence of PPIs on cytokine production. A total of 1 × 10^6^ PBMCs were cultured in 1 mL medium in a 24-well plate and preincubated with the proton-pump inhibitor (PPI) pantoprazole (75 μM) for 48 h (gray bars) or left untreated (black bars). Samples were then either stimulated with 1 μg/mL PHA (**A**,**B**,**E**,**F**) or 50 ng/mL TSST (**C**,**D**,**G**,**H**) or were left untreated for 24 h or 48 h, as indicated. The IFN-γ and IL-2 production was measured by ELISA. Experiments were performed n = 3–4 times. Results are presented as mean values ± SEM. * *p* ≤ 0.05, ** *p* ≤ 0.01 (Student’s *t*-test).

**Figure 2 ijms-24-01191-f002:**
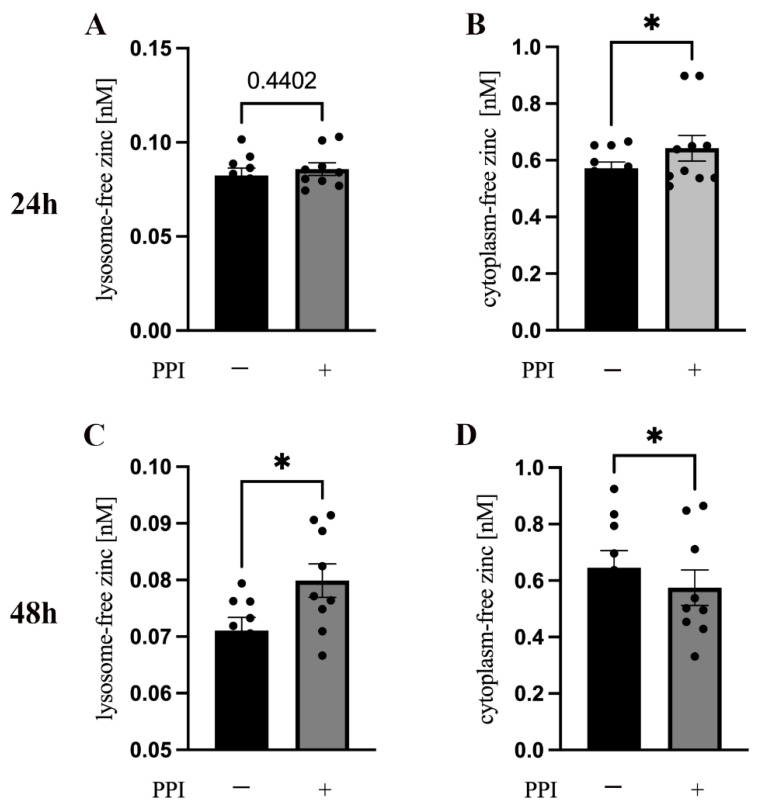
Influence of PPIs on the distribution of intracellular free zinc**.** A total of 1 × 10^6^ PBMC were cultured in 1 mL medium in a 24-well plate preincubated with the proton-pump inhibitor (PPI) pantoprazole (75 μM) (gray bars) or left untreated (black bars) for 24 h (**A**,**B**) or 48 h (**C**,**D**). Lysosome-free zinc concentration (**A**,**C**) was determined by FluoZin-3 AM, and cytoplasm-free zinc concentration (**B**,**D**) was measured using Zinpyr-1 for staining (n = 9–11). Results are presented as mean values ± SEM. * *p* ≤ 0.05 (Student’s *t*-test).

**Figure 3 ijms-24-01191-f003:**
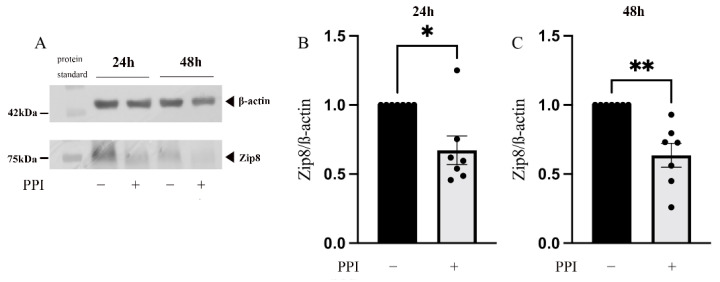
Influence of PPIs on the expression of Zip8. A total of 2 × 10^6^ PBMC/mL were incubated with the proton-pump inhibitor (PPI) pantoprazole (75 µM, gray bars) for 24 h (**B**) or 48 h (**C**) or left untreated (black bars), respectively. The expression of Zip8 was quantified by Western blot analysis. (**A**) One representative experiment out of n = 6–7 independent experiments is displayed. Results are presented as mean values ± SEM. * *p* ≤ 0.05, ** *p* ≤ 0.01 (Student’s *t*-test).

**Figure 4 ijms-24-01191-f004:**
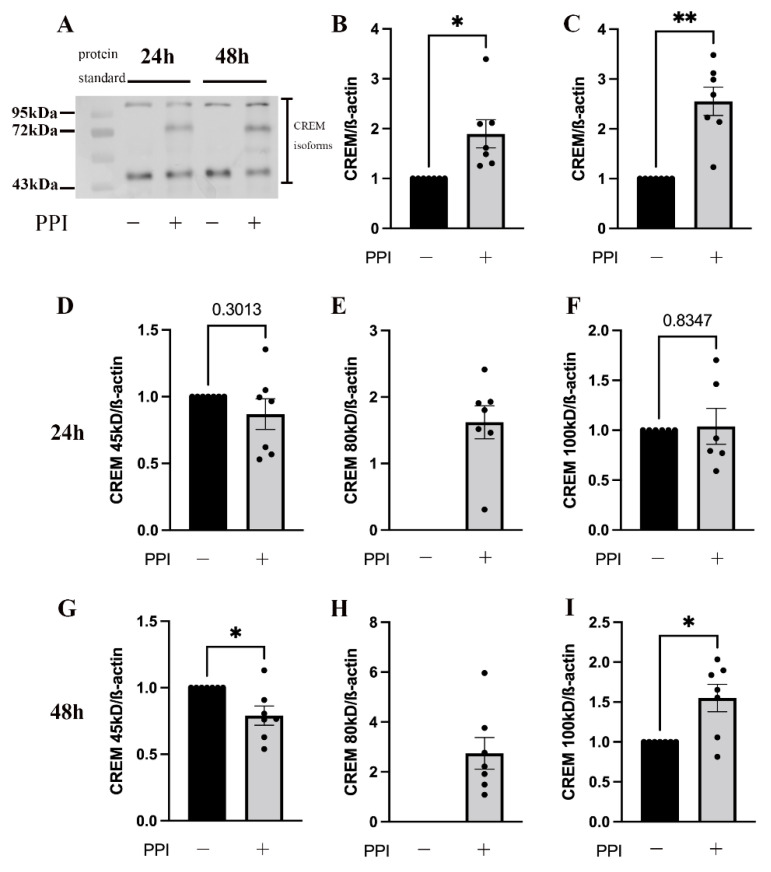
Influence of PPI on the expression of CREMα. A total of 2 × 10^6^ PBMC/mL were incubated with the proton-pump inhibitor (PPI) pantoprazole (75 μM, gray bars) for 24 h (**B**,**D**,**E**,**F**) or 48 h (**C**,**G**,**H**,**I**) or left untreated (black bars), respectively. The expression of CREMα was quantified by Western blot analysis. β-actin is the same as in Figure 3. (**A**) One representative experiment out of n = 6–7 independent experiments is displayed. Results are presented as mean values ± SEM. * *p* ≤ 0.05, ** *p* ≤ 0.01 (Student’s *t*-test).

**Figure 5 ijms-24-01191-f005:**
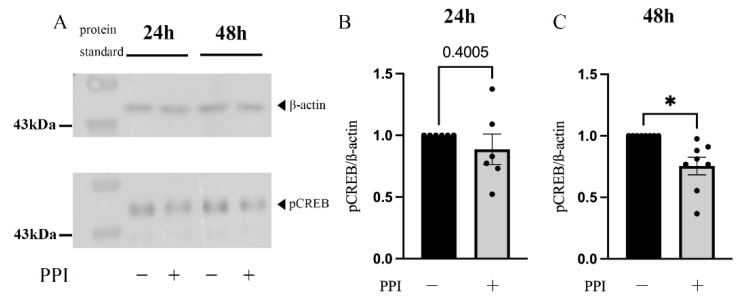
Influence of PPIs on the expression of pCREB. A total of 2 × 10^6^ PBMC/mL were incubated with the proton-pump inhibitor (PPI) pantoprazole (75 μM, gray bars) for 24 h (**B**) or 48 h (**C**) or left untreated (black bars), respectively. pCREB was quantified by Western blot analysis. (**A**) One representative experiment out of n = 6–8 independent experiments is displayed. Results are presented as mean values ± SEM. * *p* ≤ 0.05 (Student’s *t*-test).

**Figure 6 ijms-24-01191-f006:**
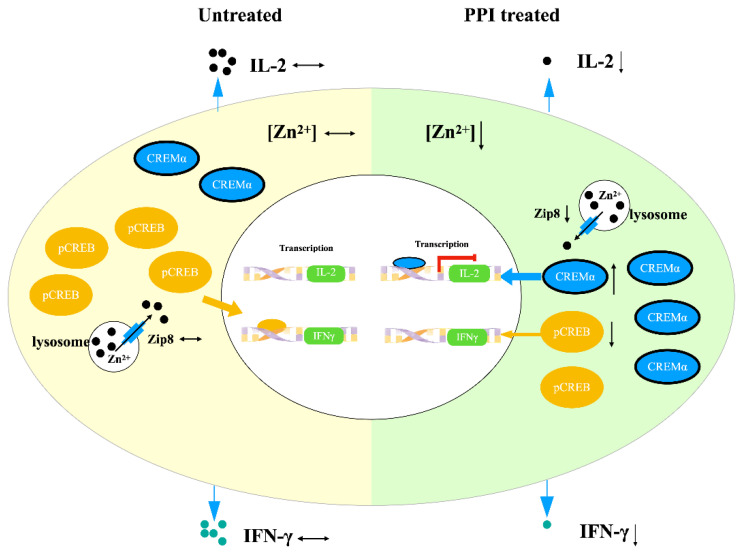
PPIs suppress T cell response by shifting intracellular zinc distribution. Proton-pump inhibitors (PPI) decrease the expression of Zip8, and thus lower the cytoplasm-free zinc concentration. This leads to the overexpression of the transcription factor CREMα and low expression of the transcription factor pCREB in peripheral blood lymphocytes. CREMα is a negative regulator of the IL-2 gene, and its overexpression dramatically limits adequate IL-2 production. pCREB is a positive regulator of the IFN-γ gene, and its low expression dramatically limits adequate IFN-γ production.

## Data Availability

The datasets generated and/or analyzed during the current study are not publicly available but are available from the corresponding author on reasonable request.

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
