# Peer review of "Proton-Pump Inhibitors Suppress T Cell Response by Shifting Intracellular Zinc Distribution"

_ijms, 2023, doi:10.3390/ijms24021191_

Round 1
Reviewer 1 Report
In the present manuscript Lui et al. studied the effect of the proton-pump inhibitor pantoprazole on T cell activation and cytokine production in vitro.
The authors could show that pantoprazole suppressed IFN-g and IL-2 production of PHA-stimulated and TSST-1-stimulated human PBMC regulating the expression of the zinc transporter Zip8 and the cAMP-responsive element modulator (CREM)a as well as the phosphorylated cAMP response element-binding protein (pCREB).
The experiments were well performed, with adequate controls. The manuscript is well written and the data provided is solid, supports their conclusions and provide an interesting and important contribution to our knowledge regarding side effects of proton-pump inhibitors as widely used drugs.
Minor points:
- Lane 83: change “48 has indicated” to “48 h as indicated”
- Figure 2 B: please adjust the gray scale
- Lane 97: please delete “for 48 h”
- Figure 3A: please enlarge the caption of the picture (PPI - + - +)
- Lane 121: change “Figure 4B-G” to “Figure 4D-I”
- Figure 3A, 4A, 5A: please add “kDa”
Author Response
Point-by-Point Reply to Reviewer 1
Reviewer:
In the present manuscript Lui et al. studied the effect of the proton-pump inhibitor pantoprazole on T cell activation and cytokine production in vitro.
The authors could show that pantoprazole suppressed IFN-g and IL-2 production of PHA-stimulated and TSST-1-stimulated human PBMC regulating the expression of the zinc transporter Zip8 and the cAMP-responsive element modulator (CREM)a as well as the phosphorylated cAMP response element-binding protein (pCREB).
The experiments were well performed, with adequate controls. The manuscript is well written and the data provided is solid, supports their conclusions and provide an interesting and important contribution to our knowledge regarding side effects of proton-pump inhibitors as widely used drugs.
Comment: We thank the reviewer for its expert evaluation and positive comments.
Reviewer Minor points:
- Lane 83: change “48 has indicated” to “48 h as indicated”
- Figure 2 B: please adjust the gray scale
- Lane 97: please delete “for 48 h”
- Figure 3A: please enlarge the caption of the picture (PPI - + - +)
- Lane 121: change “Figure 4B-G” to “Figure 4D-I”
- Figure 3A, 4A, 5A: please add “kDa”
Comment: We thank the reviewer for this hints and corrected the manuscript in all points mentioned.
Reviewer:
English language and style are fine/minor spell check required
Comment:
Manuscript was corrected by a professional proofreader.
Reviewer 2 Report
The authors complete a great literature review supporting the lines of study. They provided appropriate justification for their study aim and hypothesis. The authors hypothesized that proton pump inhibitors alter zinc homeostasis and thus affect cytokine expression.
Based on the data presented in the article there are significant increases in IFN-gamma and IL-2 in response to both PHA and TSST, inducers of T-Cell activation. Furthermore, these significant increases are attenuated by pre-incubation with PPI, suggesting a blunted immune response resultant of PPI application. The authors investigate intracellular zinc localization as a means of identifying a potential role for the micronutrient in the immune response.
The authors support their line of questioning with the reason that increased cytoplasmic free zinc concentrations inhibit activity of the phosphatase calcineurin. This inhibition of this phosphatase allows for sustained level of p-CREB which acts as a transcription factor to increase IFN-gamma expression. In Figure 2 free zinc concentrations are compared between the cytosol and the lysosomes with and without PPI treatment after 24 and 48 hours. At 24 hours there was no change in lysosomal zinc concentrations whereas there was an increase in cytoplasmic zinc concentrations. The authors provide plausible explanations for this finding. After 48 hours there is a clear decrease in cytosolic zinc and an increase lysosomal zinc. The authors demonstrate a decrease in the expression of Zip8 (figure 3), responsible for transporting zinc from the lysosome to the cytoplasm.
IFN-gamma: The authors imply that decrease in cytosolic zinc leads (figure 2) to sustained activity of calcineurin, which functions as a phosphatase for CREB-P (figure 5) and thereby reducing expression of IFN-gamma (figure 1). The authors do not show measurement of Calcineurin to substantiate this reasoning. This reviewer suggests an assay to measure to calcineurin as an additional assay to support the conclusions drawn with this paper.
IL-2: The authors report that an upregulation of CREM isoforms of 80kD and 100kD (figure 4) negatively regulate IL-2 transcription (figure 1).
Overall, the authors create a clear pathway for the reduction of both IFN-gamma and IL-2. This reviewer would like to see evidence of decrease CN activity.
Author Response
Point-by-Point Reply to Reviewer 2
Reviewer:
The authors complete a great literature review supporting the lines of study. They provided appropriate justification for their study aim and hypothesis. The authors hypothesized that proton pump inhibitors alter zinc homeostasis and thus affect cytokine expression.
Based on the data presented in the article there are significant increases in IFN-gamma and IL-2 in response to both PHA and TSST, inducers of T-Cell activation. Furthermore, these significant increases are attenuated by pre-incubation with PPI, suggesting a blunted immune response resultant of PPI application. The authors investigate intracellular zinc localization as a means of identifying a potential role for the micronutrient in the immune response.
The authors support their line of questioning with the reason that increased cytoplasmic free zinc concentrations inhibit activity of the phosphatase calcineurin. This inhibition of this phosphatase allows for sustained level of p-CREB which acts as a transcription factor to increase IFN-gamma expression. In Figure 2 free zinc concentrations are compared between the cytosol and the lysosomes with and without PPI treatment after 24 and 48 hours. At 24 hours there was no change in lysosomal zinc concentrations whereas there was an increase in cytoplasmic zinc concentrations. The authors provide plausible explanations for this finding. After 48 hours there is a clear decrease in cytosolic zinc and an increase lysosomal zinc. The authors demonstrate a decrease in the expression of Zip8 (figure 3), responsible for transporting zinc from the lysosome to the cytoplasm.
IL-2: The authors report that an upregulation of CREM isoforms of 80kD and 100kD (figure 4) negatively regulate IL-2 transcription (figure 1).
Comment: We thank the reviewer for its expert evaluation and positive comments.
Reviewer:
IFN-gamma: The authors imply that decrease in cytosolic zinc leads (figure 2) to sustained activity of calcineurin, which functions as a phosphatase for CREB-P (figure 5) and thereby reducing expression of IFN-gamma (figure 1). The authors do not show measurement of Calcineurin to substantiate this reasoning. This reviewer suggests an assay to measure to calcineurin as an additional assay to support the conclusions drawn with this paper.
Overall, the authors create a clear pathway for the reduction of both IFN-gamma and IL-2. This reviewer would like to see evidence of decrease CN activity.
Comment: We thank the reviewer for this hint. However, we just discuss that the reduction in pCREB may be due to CN inhibition and cited reference 18 for that. In contrast, Xia et al. (Metallomics 2018), showed that CN is only inhibited at much higher concentrations over 200µM, which are quite unphysiological. We included this as new reference 44. In fact, pCREB is regulated by a number of zinc-modulated signalling pathways, others and we recently described, e.g. PKA and ERK (von Bülow, J. Immunol. 2005), Ras/Raf (von Bülow, J. Immunol. 2007) and Akt via PTEN inhibition (Plum et al., Metallomics, 2014). Since all of them may contribute to this effect, we did not analyse any of these pathways, but showed that pCREB is the important mediator.
Lastly, these experiments cannot be done within the revision time of 5 days. But we are willing to do, if this is strictly necessary, although it would just repeat the experiments already done by others.
Reviewer:
English very difficult to understand/incomprehensible
Comment:
Manuscript was corrected by a professional proofreader.